# Diet-Treated Gestational Diabetes Mellitus Is an Underestimated Risk Factor for Adverse Pregnancy Outcomes: A Swedish Population-Based Cohort Study

**DOI:** 10.3390/nu14163364

**Published:** 2022-08-16

**Authors:** Inga Rós Valgeirsdóttir, Ulf Hanson, Erik Schwarcz, David Simmons, Helena Backman

**Affiliations:** 1Department of Obstetrics and Gynecology, Faculty of Medicine and Health, Örebro University, SE-701 82 Örebro, Sweden; 2Department of Women’s and Children’s Health, Uppsala University, SE-751 85 Uppsala, Sweden; 3Department of Obstetrics and Gynecology, Örebro University Hospital, Region Örebro County, PO Box 1613, SE-701 16 Örebro, Sweden; 4Department of Internal Medicine, Faculty of Medicine and Health, Örebro University, SE-701 82 Örebro, Sweden; 5School of Medicine, Western Sydney University, Campbelltown, NSW 2560, Australia

**Keywords:** diabetes in pregnancy, gestational diabetes, diet therapy, insulin, pregnancy outcomes

## Abstract

In Sweden, diet-treated gestational diabetes mellitus (GDM) pregnancies have been managed as low risk. The aim was to evaluate the risk of adverse perinatal outcomes among women with diet-treated GDM compared with the background population and with insulin-treated GDM. This is a population-based cohort study using national register data between 1998 and 2012, before new GDM management guidelines and diagnostic criteria in Sweden were introduced. Singleton pregnancies (*n* = 1,455,580) without pregestational diabetes were included. Among 14,242 (1.0%) women diagnosed with GDM, 8851 (62.1%) were treated with diet and 5391 (37.9%) with insulin. In logistic regression analysis, the risk was significantly increased in both diet- and insulin-treated groups (vs. background) for large-for-gestational-age newborns, preeclampsia, cesarean section, birth trauma and preterm delivery. The risk was higher in the insulin-treated group (vs. diet) for most outcomes, but perinatal mortality rates neither differed between treatment groups nor compared to the background population. Diet as a treatment for GDM did not normalize pregnancy outcomes. Pregnancies with diet-treated GDM should therefore not be considered as low risk. Whether changes in surveillance and treatment improve outcomes needs to be evaluated.

## 1. Introduction

The prevalence of gestational diabetes mellitus (GDM), now defined as hyperglycemia first detected during pregnancy, rather than overt diabetes [1], ranges from one to >30% globally depending on ethnicity, screening methods and the diagnostic criteria used [2].

Untreated hyperglycemia during pregnancy has a positive linear association with increasing frequency of adverse pregnancy outcomes, e.g., hypertensive disease, macrosomia, cesarean section (CS), shoulder dystocia, birth trauma, preterm delivery, large-for-gestational-age (LGA) infants and neonatal hypoglycemia [3,4]. Although no obvious blood glucose threshold has been shown for increased risk, treating hyperglycemia reduces pregnancy complications [5,6,7], with the first step being lifestyle advice including diet and exercise (“diet-treated GDM”) [8,9]. National guidelines regarding specific dietary advice for GDM do not exist in Sweden. General dietary advice for women with GDM, given either by the midwife, diabetes nurse or dietician, includes spreading food intake evenly throughout the day and eating from all food categories according to the plate method, as shown in Figure 1. Treatment with additional insulin (“insulin-treated GDM”) occurs when the blood glucose rises above the target range despite lifestyle advice.

Women with insulin-treated GDM have been considered a group at high risk of adverse perinatal outcomes. Such women have usually been managed in specialized outpatient antenatal care units in Sweden and subject to more intensified supervision, including more frequent ultrasound examinations and possibly induction of labor due to increased fetal growth. On the other hand, women with diet-treated GDM with adequate glycemic control during the study period (fasting < 6 mmol/L and <9 mmol/L 1.5 h after meal) have been considered to be at low risk of pregnancy complications, and in most places in Sweden, no extra fetal surveillance and/or interventions have been offered. This allocation to different obstetric approaches to care by diabetes treatment was not reported to have occurred in either of the two main randomized controlled trials of GDM management [6,10].

In 2013, the World Health Organization introduced new recommendations for the diagnosis of GDM [1]. These include lower glucose criteria than previously used in Sweden, which, coupled with a selective testing approach, had previously resulted in a low prevalence of GDM of ~1%. In 2015, the National Board of Health and Welfare in Sweden recommended adopting these recommendations [11]. The implementation of these new criteria varies largely in different regions in Sweden, as well as management guidelines for GDM. With the new recommended diagnostic criteria for GDM, much lower levels of hyperglycemia are treated with different lower treatment goals than previously used, and there has been a discussion on whether diet as a treatment for GDM is normalizing outcomes, and whether treating lower levels of hyperglycemia is warranted. To be able to evaluate the effects of treating different levels of hyperglycemia during pregnancy, the model of care for women with diet-treated GDM, including the glycemic threshold for the introduction of pharmacological therapy, may need to be reviewed.

The overall purpose of the study was therefore to compare pregnancy outcomes for women with diet-treated GDM with the background population and insulin-treated GDM.

## 2. Materials and Methods

This was a national register-based cohort study including all singleton births recorded in the Swedish Medical Birth Register (SMBR) between 1998 and 2012. Women with a pregestational diagnosis of type one and type two diabetes mellitus were excluded, as well as women with registered extreme values for maternal height and weight (weight < 35 or >200 kg and/or height < 140 and >200 cm). It is mandatory for each healthcare provider to report information obtained from medical records from prenatal, delivery and neonatal care to this national register. The National Board of Health and Welfare maintains the register. Only a small percentage of all newborns is missing completely (1–5% per year) [12]. Coverage, agreement and internal validity have been deemed very good for most variables [13,14,15]. All patient information is coded in the SMBR. Information on all hospital births is gathered prospectively and includes maternal demographic date, reproductive history and complications during pregnancy, delivery and the neonatal period. The SMBR does not contain data on laboratory analyses, such as blood glucose or the date of diagnosis [12].

Screening and diagnostic methods for GDM differed between regions in Sweden during the study period [16,17]. The main screening strategy for GDM was based on repeated capillary random blood glucose ≥ 8.0 mmol/L (plasma glucose ≥ 9.0 mmol/L) in combination with specified risk factors during pregnancy. These included body mass index (BMI) ≥ 30–35 kg/m^2^, non-European ethnicity, family history of type two diabetes mellitus, previous GDM or macrosomic infant (defined as birthweight ≥ 4500 g). Rarely, an OGTT was ordered if all of these risk factors were negative but either accelerated fetal growth or polyhydramnios was suspected.

The diagnosis of GDM (according to the 10th revision of the International Classification of Diseases (ICD-10)) was made with a 75-g OGTT in all parts of Sweden (using capillary blood samples). ICD codes for all variables used are defined in Appendix A.

Diagnosis of GDM was made if fasting glucose was ≥7.0 mmol/l and/or two-hour cut-off values were either ≥8.9 mmol/L, ≥10.0 mmol/L or ≥12.2 mmol/L [18]. Each region determined their own diagnostic criteria. However, over 95% of the regions used a two-hour cut-off value ≥ 10.0 mmol/L [19]. Although no national guidelines existed for escalation of therapy, which was therefore not uniform, local guidelines generally recommended insulin treatment initiation when blood glucose concentrations were above the target (fasting blood glucose ≥ 6.5 mmol/L and/or postprandial ≥ 7.0–8.0 mmol/L 1.5–2 h post meal) three times or more in one week [20]. Oral medications for treating GDM were not used during the study period.

The definition of each core outcome was made according to proposed coding and definition when attainable [21,22]. Maternal age at delivery was registered in years and pregnancy duration registered in days (according to ultrasound dating). Parity was registered as nulliparous or multipara based on the number of previous deliveries. BMI was registered at first prenatal visit and classified according to the World Health Organization definition: underweight BMI < 18.5 kg/m^2^, normal weight BMI 18.5–24.9 kg/m^2^, overweight BMI 25.0–29.9 kg/m^2^, obese class I BMI 30.0–34.9 kg/m^2^ and obese class II-III BMI ≥ 35.0 kg/m^2^. Ethnicity was based on country of birth. Nordic origin was defined as Sweden, Denmark, Finland, Norway and Iceland, and non-Nordic was constituted by individuals from all other countries. Smoking was defined as smoking (yes/no) at the time of first antenatal visit.

Chronic hypertensive disease was defined as hypertension diagnosed before pregnancy or blood pressure ≥ 140/90 mmHg before the 20th week of gestation. Gestational hypertension was defined as hypertension without proteinuria after the 20th week of gestation. Preeclampsia was defined as mild and severe. Preeclampsia overall included both mild and severe preeclampsia as well as diagnostic codes involving HELLP (hemolysis, elevated lever enzymes and low platelets) syndrome and unspecified preeclampsia. CS included both acute and elective surgery and was independent of stage of labor. Induction of labor included all trials of labor starting with induction independent of delivery mode. Assisted vaginal delivery only included vacuum extraction. Shoulder dystocia was defined based on ICD codes. Anal sphincter injury (third- and fourth-degree perineal laceration) was defined based on all vaginal deliveries and CS excluded. Preterm delivery was defined as delivery up to 258 days (≤36 weeks and 6 days). Fetal macrosomia was defined as absolute birthweight ≥ 4000 g, ≥4500 g or ≥5000 g. LGA was reported based on two different criteria: birthweight > 90th percentile and birthweight two standard deviations above the mean (both according to gestational age and sex). The latter is used in clinical practice in Sweden. Small for gestational age was reported based on two different criteria: birthweight < 10th percentile and birthweight two standard deviations below the mean (according to gestational age and sex). Reference percentiles and infant size at birth were based on data from all live-born singletons without malformations within the dataset. Intrauterine fetal death was defined according to ICD-10 codes and perinatal mortality as stillbirth or early neonatal death at ≤6 days of age (all malformations excluded). Asphyxia was defined as severe asphyxia with Apgar score < 4 at 5 min of age. Neonatal hypoglycemia was defined according to ICD-10 codes. Birth trauma was defined as spinal cord injury, peripheral nerve or brachial plexus injury, basal skull fracture or depressed skull fracture, clavicular fracture, long bone fracture (humerus, radius, ulna, femur, tibia, fibula) or cranial hemorrhage (subdural or intracerebral of any kind).

When performing the statistical analyses, continuous variables were given as mean with standard deviation (SD) when normally distributed and median with interquartile range when non-normally distributed. Categorical variables were given as a proportion (percentage). Maternal and neonatal characteristics and outcomes for women with both diet- and insulin-treated GDM were compared to the background population using Student’s unpaired *t*-test (mean for non-skewed continuous variables), Mann–Whitney U-test (median for skewed continuous variables) or Pearson’s chi-square test (categorical variables). A *p*-value of <0.05 was considered significant. Crude odds ratios with 95% confidence intervals and significance values were presented in relation to the background population (see Appendix A). Logistic regression analyses were used to adjust for potential confounders (BMI, age, smoking, parity, country of birth and chronic hypertensive disease) and determine the adjusted odds ratio with 95% confidence intervals for maternal and neonatal outcomes. Odds ratio was chosen as the prevalence numbers were low and the cohort was large. The background population used as a reference group was defined as women without GDM, type one or type two diabetes mellitus. Statistical analyses were performed using IBM SPSS statistics 25.

## 3. Results

During the study period of 1998–2012, there were 1,455,580 singleton pregnancies (without type one and type two diabetes mellitus) in Sweden, including 14,242 (1.0%) pregnancies with GDM. Of these, 8851 (62.1%) had diet-treated GDM and 5391 (37.9%) had insulin-treated GDM.

Table 1 shows that when compared to the background population, women in both diet- and insulin-treated GDM groups were older, had higher BMI, were more often multiparous and of non-Nordic origin. Smoking was more common in the insulin-treated GDM group, compared to the background population, while chronic hypertension was similar in both groups: both more common than in the background population. There were significant differences between insulin- and diet-treated GDM in all characteristics, except for neonatal sex (Appendix A).

Pregnancy duration was significantly shorter in both treatment groups, with a median of 277 days (270–284) in the diet group and 273 days (266–280) in the insulin group, compared to the median of 280 days (273–287) in the background population (*p* < 0.001, respectively). Despite this, the mean birthweight of the neonate was significantly higher in both groups: 3590 g (±629), in the diet group and 3685 g (±681) in the insulin group, compared to 3546 g (±567) in the background population (*p* < 0.001, respectively).

Gestational hypertension, preeclampsia (both mild and severe), CS and shoulder dystocia were significantly more common in both GDM groups (Table 2). Table 3 shows that birth trauma, preterm delivery before 37 weeks gestation, neonatal hypoglycemia, macrosomia and LGA infants were significantly more common in both GDM groups. There was no significant difference in intrauterine death or perinatal mortality, neither between the two treatment groups nor compared to the background population in either group (Table 3). Insulin-treated GDM generally resulted in more severe negative outcomes, compared to the diet-treated group.

## 4. Discussion

In this Swedish population-based study, adverse pregnancy outcomes among women with GDM, using relatively high diagnostic criteria, were increased regardless of whether treated with insulin or diet. The outcomes were generally worse in the insulin-treated group. Neither diet- nor insulin-treated GDM was associated with increased risk of intrauterine death or perinatal mortality compared to the background population.

Women with diet-treated GDM were almost at a twofold higher risk of having LGA neonates compared to the background population, and women with insulin-treated GDM almost at a fourfold higher risk. The neonates in both groups were at similarly increased risk for birth trauma. Macrosomia prevalence irrespective of definition was higher in the insulin-treated group, but diet-treated women had a rate of 6.3% macrosomia based on the Swedish clinically used definition of 4500 g, compared with 3.8% in the background population and 9.8% in the insulin-treated group. These results show that birthweight was not normalized in both groups, and the undiagnosed GDM in the background population most likely resulted in the underestimation of this difference. Maternal outcomes such as preeclampsia, CS and induction of labor were also significantly more common in the insulin-treated group compared to the diet group.

A major strength of this study is the inclusion of data covering almost all pregnancies in Sweden over the time period. A further strength is that the data are based on a national health register that has been validated, with quality considered good and large numbers enabling the analysis of uncommon outcomes such as perinatal mortality [14]. We were also able to account for known confounders such as BMI.

There are, however, limitations. Since this was a register-based study, it was not possible to apply all the defined core outcome sets for GDM [22], due to data that were not collected (adherence to the intervention) or were unreliable (gestational weight gain). The register does not contain data on blood glucose or other laboratory measurements. No data are recorded in the register on the timing of diagnosis and treatment, availability of dieticians, insulin dosage or adherence to the intervention and surveillance. This makes it difficult to estimate the effectiveness of the treatment (both diet and insulin) and whether each woman received optimal care. How women change their diet is not registered, and this makes it difficult to evaluate whether changes in dietary habits have an effect on glucose values and pregnancy outcomes. Many women do not achieve the glucose targets, and the proportion of glucose monitoring results above the target correlates with increasing adverse outcomes [23]. The dietary intervention might not have been enough to achieve targeted glycemic control, and treatment might also have been started too late, since the screening time point during the study period was between gestational weeks 28 and 32 in most centers. Early GDM (< 20 gestational weeks) is uncommon, since early screening usually occurs with an indication of a pregnancy previously affected by GDM. In addition, regional differences probably existed in the criteria for initiating insulin treatment and how to adjust the dosage, as well as the approaches to both screening and GDM diagnosis, as previously mentioned [19]. On the other hand, these different clinical routines did not change considerably over the study period.

Several meta-analyses have demonstrated at least partial benefit in treating GDM [7], but no randomized controlled trials have compared lifestyle and pharmacotherapy in those above the threshold for treatment intensification. Prior studies have mostly reported adverse outcomes in smaller cohorts, often using different diagnostic criteria, complicating both interpretation and comparison between studies. Despite small cohorts, a few cohort studies have been able to compare the results of adverse outcomes of pregnancies complicated with diet- and/or insulin-treated GDM to the background population. Our results are in accordance with these, showing that if the diagnostic criteria and the treatment targets being used are high, the risk of LGA still remains high for both diet- and insulin-treated groups and should not be ignored [24].

In a Finnish study, where screening strategy changes were evaluated (risk factor-based vs. comprehensive screening), the GDM prevalence only increased by 2%, but the proportion of diet-treated GDM increased when comprehensive screening was used. They also had a significant decrease in mean birthweight in the diet-treated group between the time periods, but, notably, the treatment targets were changed over the study period, making the effect of diet as a treatment unclear [25]. In a setting with a comprehensive screening strategy and lower diagnostic criteria, the rates of LGA can be reduced significantly for GDM overall [25,26,27] and in the diet-treated group perhaps even normalized compared to the background population [28]. Further, the risk of infants being born small for gestational age does not seem to increase despite lower diagnostic criteria and glucose targets [26].

The low prevalence of pregnancies complicated by GDM in this study can be explained by the general use of selective screening and high thresholds for GDM diagnostic criteria in Sweden. Changes were, however, introduced after the publication of the 2015 National Board of Health guidelines on diagnostic criteria, and it will be interesting to compare our findings with those from future studies using these lower criteria. There is also increasing interest in whether women who have mild hyperglycemia early in pregnancy should be treated and whether this can decrease their risk for adverse outcomes (i.e., if current strategies commence therapy too late) [29].

Most regions in Sweden have organized the care of diet-treated GDM within regular/routine maternal healthcare units, indicating a possibility that their risks approximate those of the background population. Many regions have had no extra surveillance or interventions during pregnancy, e.g., fetal cardiotocography, ultrasound or induction of labor before gestational week 42 if no other indications beyond GDM exist, as long as blood glucose concentrations remain within range. This is reflected by the mean pregnancy duration among diet-treated women being only a few days shorter than the background population, despite their higher rate of pregnancy complications. Our findings suggest that this model of care in Sweden, along with the overall GDM management (e.g., glucose targets), needs to be reviewed. Dietary interventions and compliance need to be integrated in quality registers used in Sweden to be able to evaluate and improve outcomes. The Swedish CDC4G (Changing Diagnostic Criteria for Gestational Diabetes) study [30], a stepped wedge national cluster randomized controlled trial, will hopefully give some answers regarding whether, with lower diagnostic criteria, a strict treatment protocol and better pregnancy surveillance, including more frequent ultrasound measurements, will give better outcomes on a population level. Similarly, the TOBOGM (Treatment of Booking Gestational Diabetes Mellitus) study [31], a multicenter randomized controlled trial (with one Swedish site) examining whether treating GDM diagnosed before 20 weeks of gestation may improve pregnancy outcomes, will also inform us about potential policy changes. More research on which blood glucose targets optimize outcomes depending on diagnostic criteria might help in adapting individual monitoring and treatment.

## 5. Conclusions

The increased risk for adverse perinatal outcomes in both diet- and insulin-treated GDM compared to the background population indicates that the commonly used management approaches in Sweden have been insufficient to normalize pregnancy outcomes. Moreover, some outcomes for diet-treated GDM are of the same degree as for the insulin-treated group, suggesting that the dietary intervention provided might not have been enough for achieving optimal glycemic control and/or that escalation to insulin therapy may not have occurred. There is a need for a change in management and surveillance, with individually adjusted treatment options, to be able to improve pregnancy outcomes. Future studies should also focus on long-term health implications for the mother and child.

## Figures and Tables

**Figure 1 nutrients-14-03364-f001:**
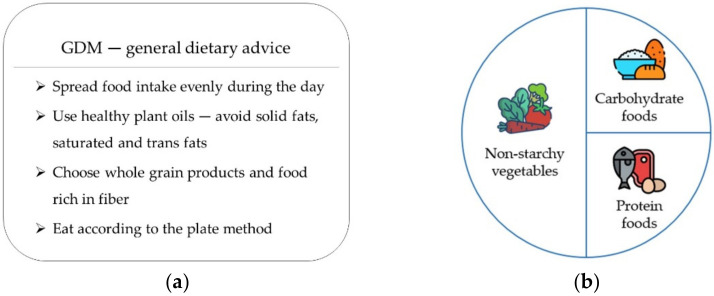
Dietary advice for women with gestational diabetes mellitus (GDM): (**a**) general dietary advice; (**b**) the plate method.

**Table 1 nutrients-14-03364-t001:** Maternal and neonatal characteristics among singleton pregnancies diagnosed with diet- or insulin-treated GDM.

	Background Population ^a^ *n* = 1,441,338	Diet vs. Background *n* = 8851		Insulin vs. Background *n* = 5391	
			** *p* ** **-Value ^b^**		** *p* ** **-Value ^b^**
** Maternal characteristics: **					
**Maternal age at delivery (years); mean ± SD**	30.0 ± 5.2	32.0 ± 5.4	<0.001	32.7 ± 5.4	<0.001
**BMI at first prenatal visit (kg/m^2^); mean ± SD**	24.5 ± 4.4	27.8 ± 6.0	<0.001	30.3 ± 6.5	<0.001
**Underweight ^c^; *n* (%)**	31,519 (2.2)	112 (1.3)	<0.001	35 (0.6)	<0.001
**Normal weight ^c^; *n* (%)**	790,384 (54.8)	2916 (32.9)	<0.001	1042 (19.3)	<0.001
**Overweight ^c^; *n* (%**	318,563 (22.1)	2425 (27.4)	<0.001	1403 (26.0)	<0.001
**Obese class I ^c^;** ** *n* ** **(%)**	101,285 (7.0)	1496 (16.9)	<0.001	1161 (21.5)	<0.001
**Obese class II-III ^c^;** ** *n* ** **(%)**	40,196 (2.8)	1006 (11.4)	<0.001	1088 (20.2)	<0.001
**Multipara; *n* (%)**	799,806 (55.5)	5400 (61.0)	<0.001	3750 (69.6)	<0.001
**Non-Nordic; *n* (%)**	275,768 (19.1)	3347 (37.8)	<0.001	2351 (43.6)	<0.001
**Smoking; *n* (%)**	117,825 (8.2)	715 (8.1)	0.741	554 (10.3)	<0.001
**Chronic hypertension; *n* (%)**	4277 (0.3)	100 (1.1)	<0.001	92 (1.7)	<0.001
** Neonatal characteristics: **					
**Male sex; *n* (%)**	741,243 (51.4)	4564 (51.6)	0.811	2818 (52.3)	0.221

GDM gestational diabetes mellitus, N number of individuals, SD standard deviation, BMI body mass index. Data are given as mean ± SD or as number of individuals and proportion N (%). ^a^ Background population as reference; pregnancies with gestational diabetes mellitus, type one and type two diabetes mellitus excluded. **^b^** *p*-Value: For continuous variables, unpaired *t*-test (maternal age, BMI) was used, with chi-square test for categorical variables (all other maternal and neonatal characteristics). ^c^ BMI classification according to the World Health Organization definition: underweight BMI < 18.5 kg/m^2^, normal weight BMI 18.5–24.9 kg/m^2^, overweight BMI 25.0–29.9 kg/m^2^, obese class I BMI 30.0–34.9 kg/m^2^ and obese class II-III BMI ≥ 35.0 kg/m^2^.

**Table 2 nutrients-14-03364-t002:** Maternal pregnancy outcomes in singleton pregnancies diagnosed with diet- or insulin-treated GDM.

	Background Population ^a^ *n* = 1,441,338		Diet vs. Background *n* = 8851				Insulin vs. Background *n* = 5391				Insulin vs. Diet	
	** *n* **	**%**	** *n* **	**%**	**aOR**	**95% CI**	** *n* **	**%**	**aOR**	**95% CI**	**aOR**	**95% CI**
**Gestational hypertension**	13,531	0.9	177	2.0	1.67	1.42–1.96	134	2.5	1.89	1.57–2.28	1.14	0.89–1.47
**Preeclampsia, overall**	37,383	2.6	500	5.6	1.71	1.55–1.88	428	7.9	2.11	1.88–2.36	1.26	1.09–1.46
**Preeclampsia, mild**	25,879	1.8	376	4.2	1.81	1.62–2.03	314	5.8	2.15	1.89–2.45	1.21	1.02–1.44
**Preeclampsia, severe**	10,079	0.7	106	1.2	1.35	1.10–1.66	95	1.8	1.77	1.42–2.22	1.35	1.00–1.83
**Cesarean section**	213,477	14.8	1853	20.9	1.18	1.11–1.25	1654	30.7	1.84	1.73–1.96	1.56	1.43–1–69
**Induction**	162,976	11.3	1643	18.6	1.40	1.32–1.48	1908	35.4	3.09	2.90–3.28	2.25	2.07–2.45
**Vacuum extraction**	102,528	7.1	587	6.6	1.02	0.93–1.11	314	5.8	1.05	0.92–1.18	1.03	0.88–1.20
**Shoulder dystocia**	2855	0.2	55	0.6	2.25	1.69–3.01	53	1.0	3.07	2.27–4.16	1.52	1.00–2.31
**Anal sphincter injury ^b^**	5041	0.4	32	0.6	1.35	0.93–1.97	18	0.6	1.49	0.89–2.48	1.08	0.57–2.04

GDM gestational diabetes mellitus, N number of individuals, aOR adjusted odds ratio, CI confidence interval. Adjusted for maternal age, body mass index, country of birth, chronic hypertensive disease, smoking and parity. ^a^ Background population as reference; pregnancies with gestational diabetes mellitus, type one and type two diabetes mellitus excluded. ^b^ All cesarean sections excluded.

**Table 3 nutrients-14-03364-t003:** Neonatal outcomes in singleton pregnancies diagnosed with diet- or insulin-treated GDM.

	Background Population ^a^ *n* = 1,441,338		Diet vs. Background *n* = 8851				Insulin vs. Background *n* = 5391				Insulin vs. Diet	
	** *n* **	**%**	** *n* **	**%**	**aOR**	**95% CI**	** *n* **	**%**	**aOR**	**95% CI**	**aOR**	**95% CI**
**Preterm delivery** **(<37 weeks)**	70,939	4.9	713	8.1	1.58	1.45–1.72	634	11.8	2.32	2.12–2.55	1.47	1.30–1.67
**Macrosomia ^b^ ≥4000 g**	278,997	19.4	2 216	25.0	1.28	1.21–1.34	1 727	32.0	1.65	1.54–1.75	1.24	1.14–1.35
**≥4500 g**	55,086	3.8	557	6.3	1.40	1.28–1.54	531	9.8	1.89	1.71–2.09	1.33	1.16–1.52
**≥5000 g**	7149	0.5	114	1.3	1.97	1.62–2.39	100	1.9	2.10	1.69–2.62	1.07	0.80–1.44
**LGA-SD ^c^**	49,013	3.4	911	10.3	2.52	2.34–2.72	1 158	21.6	5.13	4.77–5.53	2.06	1.86–2.29
**LGA-90 ^d^**	139,811	9.7	1 666	18.8	1.85	1.75–1.96	1 748	32.4	3.38	3.17–3.60	1.80	1.65–1.96
**SGA-SD ^e^**	33,544	2.3	200	2.3	0.85	0.73–0.99	89	1.7	0.58	0.46–0.73	0.72	0.54–0.95
**SGA-10 ^f^**	148,846	10.3	835	9.4	0.88	0.81–0.95	308	5.7	0.53	0.46–0.60	0.63	0.54–0.73
**Intrauterine death ^g^**	4509	0.3	32	0.4	0.97	0.68–1.38	18	0.3	0.70	0.43–1.15	0.76	0.41–1.40
**Perinatal mortality ^h^**	5925	0.4	37	0.4	0.86	0.62–1.20	24	0.5	0.71	0.46–1.09	0.89	0.52–1.51
**Apgar score <4 at 5 min.**	4943	0.3	42	0.5	1.17	0.86–1.59	48	0.9	1.71	1.25–2.34	1.46	0.94–2.27
**Hypoglycemia**	32,274	2.2	1 774	20.0	9.62	9.08–10.19	1 590	29.5	14.54	13.60–15.55	1.57	1.44–1.70
**Birth trauma**	2809	0.2	42	0.5	1.88	1.37–2.58	40	0.7	2.26	1.61–3.20	1.28	0.80–2.04
**Hyperbilirubinemia**	51,522	3.6	492	5.6	1.42	1.29–1.57	408	7.6	1.89	1.69–2.11	1.32	1.14–1.53
**Respiratory distress**	38,042	2.6	303	3.4	1.20	1.06–1.36	331	6.1	2.15	1.91–2.43	1.81	1.52–2.15

GDM gestational diabetes mellitus, N number of individuals, aOR adjusted odds ratio, CI confidence interval, LGA large for gestational age, SD standard deviation, SGA small for gestational age. Adjusted for maternal age, body mass index, country of birth, chronic hypertensive disease, smoking and parity. ^a^ Background population as reference; pregnancies with gestational diabetes mellitus, type one and type two diabetes mellitus excluded. ^b^ Macrosomia definition according to different cut-offs. ^c^ Birthweight two standard deviations above the mean, according to gestational age and sex. ^d^ Birthweight >90th percentile. ^e^ Birthweight two standard deviations below the mean, according to gestational age and sex. ^f^ Birthweight <10th percentile. ^g^ All malformations excluded. ^h^ Stillbirth or early neonatal death at ≤6 days of age, all malformations excluded.

## Data Availability

The data presented in this study are available from the corresponding author on reasonable request and according to local ethical and policy regulations.

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
