# Peer review of "Diet-Treated Gestational Diabetes Mellitus Is an Underestimated Risk Factor for Adverse Pregnancy Outcomes: A Swedish Population-Based Cohort Study"

_nutrients, 2022, doi:10.3390/nu14163364_

Round 1

Reviewer 1 Report

The article presents originality as very few studies compare the effect of gestational diabetes treatment with diet versus insulin on maternal and fetal outcomes. The results obtained are from a cohort, with a large number of pregnant women assessed. Furthermore, the results were adjusted for several confounding variables, namely BMI, hypertension, etc. However, there is one very important piece of information missing, the glycaemic control of pregnant women throughout pregnancy in both groups. It is known that regardless of treatment, glycaemic control will have a profound impact on maternal and foetal complications associated with gestational diabetes. Thus, when the authors state that pregnant women on diet treatment had some of the complications at similar levels to women on insulin treatment, the outcome may be biased by poor glycaemic control through diet alone (and a better control achieved by insulin treatment). Therefore, and considering that it is not possible to collect this information at the moment, this limitation in the interpretation of the results obtained should be more clearly defined in the article.

Minor suggestions:

Lines 178 and 179 “while chronic hypertension was similar in both groups.”

Table 1: it would be interesting to compare maternal and neonatal characteristics from diet versus insulin treated groups

Regarding macrossomia, in the results it appears birth weight above 4000g and in the discussion above 5000g

Reviewer 2 Report

This is the report of a retrospective data-mining exercise using the Swedish Medical Birth Register in 15-year period pre-dating stricter recommendations for diagnosis from WHO based on studies such as HAPO. It is a major undertaking. Unfortunately, the outcomes are extremely limited. The authors/investigators faced several challenges for which they cannot compensate. Clinical investigation and management varied widely across different geographic areas of Sweden. This of itself adds a complexity and heterogeneity which cannot be controlled and is unknown to the investigators.

2.      The data base does not record any glucose concentration data, apart from stating the diagnosis of GDM is in a subgroup of women (variously selected as noted above) and in whom 75 g OGTTs are undertaken. The glucose criteria for diagnosis are very high, and essentially are for any Diabetes mellitus diagnosis. The result is an apparent incidence of GDM 0f 1%.

3.      The interventions used in this group are diet or insulin. However, there are no data available for the investigators to know when those treatment were introduced or whether they were exclusive; i.e. could Diet be converted to Insulin if control was not satisfactory. But again, there are no indications as to what the targeted control concentration or range were.  

 Considering specifically the paper submitted. The women who underwent OGTT initially are a select subset, pre-ordained by consideration of ethnicity, previous history etc. It would have been interesting to know how many OGTT were undertaken to give the study population of approximately 14,000. The reported 19% incidence of macrosomia in the normal population is high compared to many other national population studies in the same period and suggestive that a significant number of women with GDM were not identified. Comment is required.

Data show a gradation from “normal” to “diet” to Insulin” across many parameters measured. And the inescapable conclusion that Diet treated GDM in this group is still high risk is consistent with that. But had the authors considered that Insulin is treatment which may have been after “failed” diet and too late into pregnancy to be effective producing measurable reduced foetal weight gain for instance.

As supporting data are not available for degree of control targeted and times of intervention are not available the length of the discussion should be foreshortened as too much speculation is involved.

Specific suggestions:

Introduction: "Adequate glycaemic control" requires definition here or elsewhere (with reference). While this paper is aimed at a Swedish audience, its findings are of wider interest and references to pratice elsewhere would be relevant.

Methods: (1) can birthing data be entered into a register prospectively? (2) Methods section as currently presented is very long and does not read easily.  I would suggest a summary with a supplementary table giving all the specific codes would be of assitance to the reader. (3) Abbreviations such as BMI, CS while common should possible be listed in an appendix or codicil. 

Results and Discussion: Most comments are included generally above. 
